# Implementation of the Onsager Theorem to Evaluate the Speed of the Deflagration Wave

**DOI:** 10.3390/e22091011

**Published:** 2020-09-10

**Authors:** Eran Sher, Irena Moshkovich-Makarenko, Yahav Moshkovich, Beni Cukurel

**Affiliations:** Faculty of Aerospace Engineering, Technion-Israel Institute of Technology, Haifa 3200003, Israel; i.makrenko@gmail.com (I.M.-M.); yahavmoshko@gmail.com (Y.M.); bcukurel@technion.ac.il (B.C.)

**Keywords:** deflagration wave, Onsager theorem

## Abstract

While considering the deflagration regime, the thermal theory of combustion proposes that the mechanism of heat transfer from the flame exothermic zone to the front neighborhood reactants layer dominates the flame behavior. The introduction of the Fourier law allows a closed solution of the continuity and energy conservation equations to yield the burning velocity. It is, however, clear that this classical solution does not conform to the momentum equation. In the present work, instead of introducing the Fourier law, we suggest the introduction of a simplified version of the Onsager relationship, which accounts for the entropy increase due to the heat transfer process from the front layer to its successive layer. Solving for the burning velocity yields a closed solution that also definitely conforms to the momentum equation. While it is realized that the pressure difference across the flame front in the deflagration regime is very small, we believe that violating the momentum equation is intolerable. Quite a good fitting, similarly to the classic theory predictions, has been obtained between our predictions and some experimentally observed values for the propagation flame deflagration velocity, while here, the momentum equation is strictly conserved.

## 1. Background

Flame propagation in a flammable quiescent homogeneous medium under steady-state, steady-flow (SSSF) conditions may occur under two different discrete regimes: deflagration (sub-sonic), or detonation (super-sonic). In 1886 Mallard and Le-Chateliet were the first to observe how a flammable gas/air mixture that is ignited at the open end of a long tube propagates and develops along the tube. Since then, a large body of experimental observations has been collected for a number of fuels under a range of initial conditions. The two different regimes were clearly noted and a number of successful attempts were made to explain them. Some useful expressions have also been developed to enable reasonable predictions.

When the three conservation equations (continuity, momentum and energy) are applied to a one-dimensional SSSF flame propagation process in a tube, it may soon be realized that in addition to a suitable equation of state, one more equation is needed to solve them [1]. The five unknowns are the inlet velocity, exit velocity, exit pressure, exit temperature and exit density. Of particular interest to us is the inlet velocity since it is defined as the basic flame propagation velocity. Rankine, Hugoniot and Rayleigh [1] showed that this set of equations (the three conservation equations combined with the equation of state) yields two simple algebraic equations (one quadratic and one linear). Customarily, the two equations are plotted on a pressure-density^−1^ coordinate system. The intersections of these two curves define the different regimes. It can be shown that the slope of the Rankine–Rayleigh equation (linear equation) depends on the flame propagation velocity, which is an unknown parameter. It was suggested by Chapman and Jouguet [1] that the two tangent points between these two curves represent the deflagration and the detonation conditions while the two different slopes of the Rankine–Rayleigh equation represent the two propagation velocities. When comparing the results with available experimental observations, it is clear that the predicted detonation propagation velocity fits well. The predicted propagation velocity in the deflagration regime is however far higher, about one or two orders of magnitude higher than the observed value. The thermal theory of combustion date back to Mallard and Le Chatelier [1], who proposed that unlike in the detonation region, where the hydrodynamic effects govern the flame propagation, when deflagration is considered, the mechanism of heat transfer from the flame exothermic zone to the front neighborhood reactants layer dominates the flame behavior. Zeldovich, Frank-Kamenetskii and Semenov [2] further improved the concept and presented it in detail. When the Fourier law is introduced to the above set of equations, the five equations allow a closed solution [2]. Some plausible assumptions may result in simple algebraic expression for the propagation velocity. It, however, can be shown that this classical solution does not conform to the momentum equation [[1], p. 150 line 42]. It is true that the pressure difference across the flame front in the deflagration regime is very small, but still, violating the momentum equation can hardly be accepted.

In the present work, we suggest a new approach to bridge this classic gap. Instead of introducing the Fourier law, we suggest introducing the Onsager relationship [3], which accounts for the entropy increase due to the heat transfer process from the front layer to its successive layer. Based on a few logical assumptions, a simplified Onsager relationship has been introduced to the set of the conservation equations, and a closed solution that definitely conforms to the momentum equation has been obtained. A very good fitting has been obtained between the predicted and the experimentally observed values of the propagation flame deflagration velocity.

## 2. Theory

Consider a combustion wave propagating in SSSF conditions through a combustible mixture in a two open-sided, open-ended adiabatic stationary tube. The wave is opposed by the reactants (unburned gases) flowing at a velocity exactly equal to the wave propagation velocity, so that the wave is fixed with respect to the tube. Following the classical theory, while neglecting any transport phenomena in the fluid (no viscosity, heat transfer and matter diffusivity), we may apply the following conservation equations between the tube entrance (reactants) and the tube exit (products):

Continuity:(1)$$\rho_{R}u_{R}=\rho_{P}u_{P}$$

Momentum:(2)$$\rho_{R}u_{R}^{2}+P_{R}=\rho_{P}u_{P}^{2}+P_{P}$$

Energy:(3)$$\frac{u_{R}^{2}}{2}+h_{R}=\frac{u_{P}^{2}}{2}+h_{P}$$
where ρ, P, and u are the fluid density, pressure and velocity, respectively, and h its enthalpy. The indices *R* and *P* indicate reactant and product, respectively. Further we assume that both the reactants and products follow the ideal gas law, thus:(4)P=ρRT

We have now four equations with five unknowns: the temperature, pressure, density and velocity of the products, and the reactants’ velocity. The classical theory suggests that if we further assume invariant gas constant across the flame and identical heat capacity for the reactants and products, Equations (1)–(5) may be easily combined to yield the following two relationships [1]:

The Rayleigh equation:(5)$$P_{P}=P_{R}-\left(\frac{1}{\rho_{P}}-\frac{1}{\rho_{R}}\right)\rho_{R}^{2}u_{R}^{2}$$

The Hugoniot equation:(6)$$\frac{k}{k-1}\left(\frac{P_{P}}{\rho_{P}}-\frac{P_{R}}{\rho_{R}}\right)-\frac{1}{2}\left(P_{P}-P_{R}\right)\left(\frac{1}{\rho_{R}}+\frac{1}{\rho_{P}}\right)=h_{c}$$
where k is the specific heat ratio and $h_{c}$ the heat of combustion. Now we have two equations with three unknowns. The Rayleigh and Hugoniot equations may be plotted on a P−1/ρ coordinate system, while the intersections of the Rayleigh lines with the Hugoniot curve (four intersection points) define four possible solutions. Note that the two different families of the Rayleigh lines slopes (two families of beams) that intersect with the Hugoniot curve signify the reactants’ velocities in the detonation and in the deflagration regimes, which are still unknown parameters. When considering the slopes of these two families, it is clear that there are only two discrete slopes that generate two tangent beams to the Hugoniot curve, the steeper beam designates the detonation wave and the milder beam designates the deflagration wave. Stability and entropy considerations reveal that the only stable detonation wave is represented by the tangent point between the Rayleigh and the Hugoniot curves (the upper Chapman–Jouguet point) [1]. The suitable algebraic equation for the tangent conditions provides the additional equation that is needed to solve Equations (1)–(4) (or (5) and (6)). A wide range of experimental observations confirmed the velocity of the detonation wave as predicted by this set of equations. The prediction of the deflagration velocity has however been found far higher (1–2 order of magnitudes) than the observed value.

It was proposed by Mallard and Le Chatelier that the structure of the deflagration wave (which is essentially different from the structure of the detonation wave) suggests that the flame propagation under the deflagration regime is controlled mainly by the heat transfer from the flame front exothermic layer to the fresh reactants’ layer ahead of it [1]. Semenov, Zeldovich, and Frank-Kamenetskii adopted this concept and considered the Fourier law of heat conduction to take account of it [2]:(7)$$J_{t}=-K_{t}\frac{dT}{dx}$$
where $J_{t}$ is the heat flux and $K_{t}$ is the thermal conductivity of the reactants.

While assuming a linear temperature profile and ignoring the mass diffusion, the classical thermal theory [1] suggests equating the heat conduction to the energy increase from reactants to products, thus obtaining an algebraic equation that provides the additional equation that is needed to solve Equations (1)–(4) (or (5) and (6)) for a deflagration wave. Further simplifications and the introduction of the flame thickness concept [1] yield the classical relationship between the laminar burning velocity, $S_{L}$ (=the reactants’ velocity), the flame thickness, δ, and the thermal diffusivity of the reactants, α:(8)$$S_{L}=\frac{\alpha }{\delta }$$

Experimental observations confirmed the velocity of the deflagration wave as predicted by Equation (8). It can be shown however that this classic simple solution does not conform to the momentum equation [[1], p. 150 line 42]. Although the pressure difference across the flame front in the deflagration regime is very small, the violation of the momentum equation is improper.

## 3. Present Suggested Approach

As an alternative to the Semenov group’s methodology, we propose in the present work a different approach to bridge this gap. Instead of introducing the Fourier law, we suggest introducing the Onsager relationship, which accounts for the entropy increase due to the heat transfer process from the front layer to its successive layer (see [3,4] for the essentials of the Onsager relationship). Based on a few logical assumptions, a simplified Onsager relationship can be derived and introduced to the set of conservation equations. We start from a basic postulation that suggests, in general, when a system is close to the state of equilibrium [4], any flux $J_{i}$ (heat, molecules, momentum, etc.) is expected to be proportional to the sum of linear functions of any driving force $X_{j}$ (gradient of temperature^−1^, concentration, velocity, etc.). Thus:(9)$$J_{i}=\sum_{j=1}^{n}L_{ij}X_{j}^{}$$
where the coefficients $L_{ij}$ are the Onsager phenomenological coefficients. Based on statistical thermodynamics, Onsager [4] showed that whenever the total entropy production per unit time per unit volume resulting from all the irreversible processes can be represented as a linear sum of products of forces and fluxes, then $L_{ij}=L_{ji}$. Thus:(10)$$\frac{dS}{dt}=\sum_{i}J_{i}X_{i}>0$$

We assume that in the case of combustion, the contribution of the molecular diffusivity to the entropy production is marginal, and thus the entropy is generated due to two pronounced processes: heat generation and heat conduction, thus:(11)$$\frac{dS}{dt}={\left(\frac{dS}{dt}\right)}_{heat\;generation}+{\left(\frac{dS}{dt}\right)}_{heat\;conduction}$$

Assuming that the main heat production occurs at the flame front at $T_{P}$, and thus the entropy production due to the heat generation (here the driving force is the gradient of 1/T, while the flux is the heat), is:(12)$${\left(\frac{dS}{dt}\right)}_{heat\;generation}=\frac{\dot{Q}}{\varepsilon \delta T_{P}}=\frac{\dot{m}h_{c}}{\varepsilon \delta T_{P}}=\frac{\rho_{R}S_{L}h_{c}}{\varepsilon \delta T_{P}}$$
where $\dot{m}$ and εδ are the mass flow rate per unit area and the reactions’ zone thickness (ε is an arbitrary number smaller but close to 1), respectively. $\dot{Q}$ is the heat production per unit area. Assuming $\frac{dT}{dx}=const$, and K=const (here K is the thermal conductivity of the gases in $\left[\frac{kJ}{m\cdot s\cdot K}\right]$ units), yields:(13)$${\left(\frac{dS}{dt}\right)}_{heat\;conduction}=K\frac{{\left(T_{P}-T_{R}\right)}^{2}}{T_{P}T_{R}\delta^{2}}$$

Assuming an ideal gas with $c_{p}=const$, and that the entropy production equals to the entropy increase from reactants to products results:(14)$$K\frac{{\left(T_{P}-T_{R}\right)}^{2}}{T_{P}T_{R}\delta^{2}}+\frac{\rho_{R}S_{L}h_{c}}{\varepsilon \delta T_{P}}=\frac{\rho_{R}S_{L}}{\delta }\left[c_{p}ln\left(\frac{T_{P}}{T_{R}}\right)-Rln\left(\frac{P_{P}}{P_{R}}\right)\right]$$

Note: the right term in Equation (14) represents the entropy increase from reactants to products. This term is obtained while integrating the thermodynamic relationship Tds=dh−vdp from the inlet to the exit, while assuming an ideal gas law with $c_{p}=const$.

Solving for $S_{L}$:(15)$$S_{L}=\frac{K}{\rho_{R}c_{p}\delta }*\frac{{\left(T_{P}-T_{R}\right)}^{2}}{T_{P}T_{R}\left[ln\left(\frac{T_{P}}{T_{R}}\right)-\frac{k-1}{k}ln\left(\frac{P_{P}}{P_{R}}\right)-\frac{h_{c}}{\varepsilon c_{p}T_{P}}\right]}$$

Equation (15) correlates the flame laminar velocity with three unknowns: $T_{P}$, $P_{P}$ and the flame thickness *δ.* Following the classic thermal theory, in which the heat generation is compared to the heat convection, the flame thickness can be easily expressed in terms of the rate of heat generation, which include an Arrhenius-type expression [1]. It is noted that the result presented in Equation (15) resembles the classical result obtained from the energy balance between heat conduction and heat convection, for which the right term = 1. Equations (1)–(4) and (15) thus form a complete set of five equations with five unknowns: the temperature, pressure, density and velocity of the products, and the reactants’ velocity, and thus conforming to the three conservation laws, which include the momentum equation.

## 4. Discussion

Some important results of the present calculations are depicted in Figure 1. This schematic chart shows the range of the two regimes (classic map), together with the variation of the products’ entropy for reactants that initially are at standard conditions. As expected, the stable deflagration wave is located on the Hugoniot curve (black curve) on the deflagration regime side $\left(1/\rho >1/\rho_{R}\right)$, where the products’ density is lower than that of the Chapman–Jouguet (C-J) lower point, while the pressure is lower than that of the reactants. The increase in the entropy at this point reflects the entropy gain due to the heat generation and the heat conduction as suggested in Equation (11).

For typical values of a stoichiometric hydrocarbon/air mixture, the right term of Equation (15), is in the order of one. Moreover, since ε is an arbitrary number that is smaller but close to one, its effect on the burning velocity (Equation (15)) is practically insignificant. For the following brief comparison, we further assume that ε≈1. The effect of the reactants’ temperature, pressure and equivalence ratio on the burning velocity are therefore mainly attributed to their effects on the flame thickness and to a smaller extent on the reactants’ density. This is in a similar manner as suggested by the classical result, which is obtained from the energy balance between heat conduction and heat convection while ignoring the pressure difference across the flame. The ratio between the predicted burning wave velocity, $S_{L}$, of the present approach and that of the classical thermal theory is:(16)$$\frac{{\left(S_{L}\right)}_{Present\;approach}}{{\left(S_{L}\right)}_{Thermal\;theory}}=\frac{{\left(T_{P}-T_{R}\right)}^{2}}{T_{P}T_{R}\left[ln\left(\frac{T_{P}}{T_{R}}\right)-\frac{k-1}{k}ln\left(\frac{P_{P}}{P_{R}}\right)-\frac{h_{c}}{\varepsilon c_{p}T_{P}}\right]}$$

While the present approach ensures that the momentum basic law is conserved, Equation (16) suggests that the end results for the burning velocity do not noticeably differ from the value as predicted by the classical theory. For a typical hydrocarbon flame, the ratio is close to one.

Figure 2 shows some experimental results and the present predictions (solid line) for methane/air mixtures. As expected, the predicted results show quite a good fitting to the observed values of several experimental data, as collected by Andrews and Bradley [5] in a wide range of initial conditions. It is however noted that since the Arrhenius-type equation for a single-step reaction does not maximize at or close to the stoichiometric conditions, the predictions for fuel rich mixtures (φ>1 in Figure 2c) are inadequate.

## 5. In Summary

The thermal theory of combustion proposes that for deflagrating flames the mechanism of heat transfer from the flame exothermic zone to the front neighborhood reactants layer dominates the flame behavior. The introduction of the Fourier law provides the additional required equation that after some plausible assumptions allows a closed solution to yield the burning velocity. It is however clear that this classical solution does not conform to the momentum equation. In the present work, instead of introducing the Fourier law, we suggest introducing a simplified version of the Onsager relationship, which accounts for the entropy increase due to the heat transfer process from the front layer to its successive layer. In this way, a closed solution that definitely conforms to the momentum equation has been obtained. It is anticipated that the pressure difference across the flame front in the deflagration regime is very small, but still, violating the momentum equation can hardly be accepted. Unsurprisingly, quite a good fitting has been obtained between the predicted and the experimentally observed values of the propagation flame deflagration velocity, while the momentum equation is strictly conserved.

## Figures and Tables

**Figure 1 entropy-22-01011-f001:**
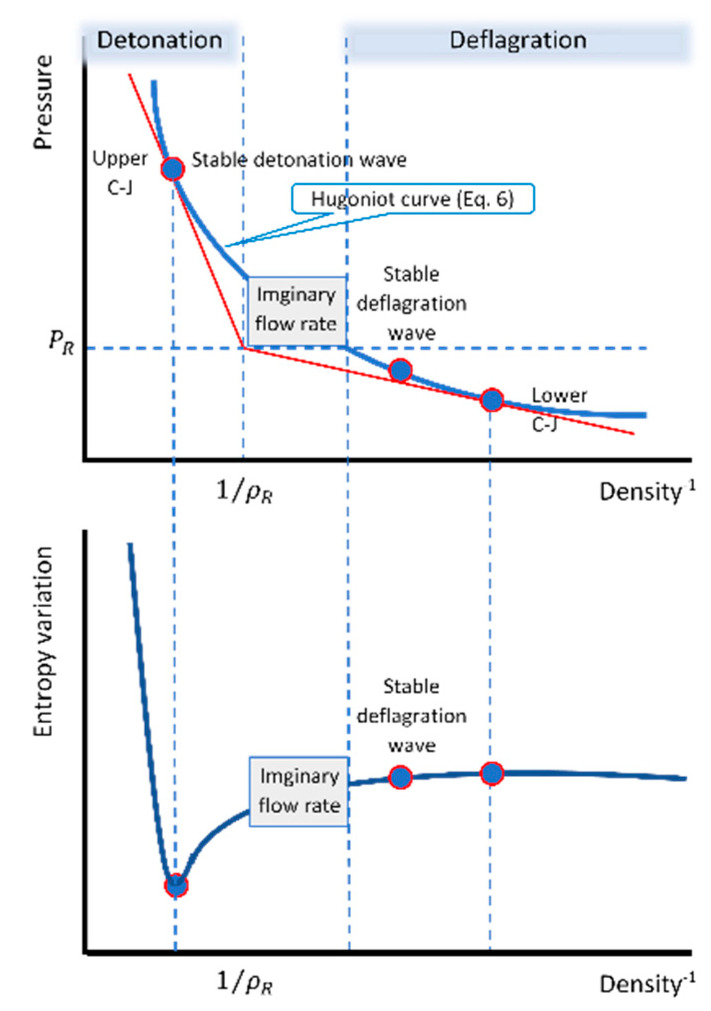
Schematic chart of the range of the two regimes (classic map), together with the variation of the products’ entropy.

**Figure 2 entropy-22-01011-f002:**
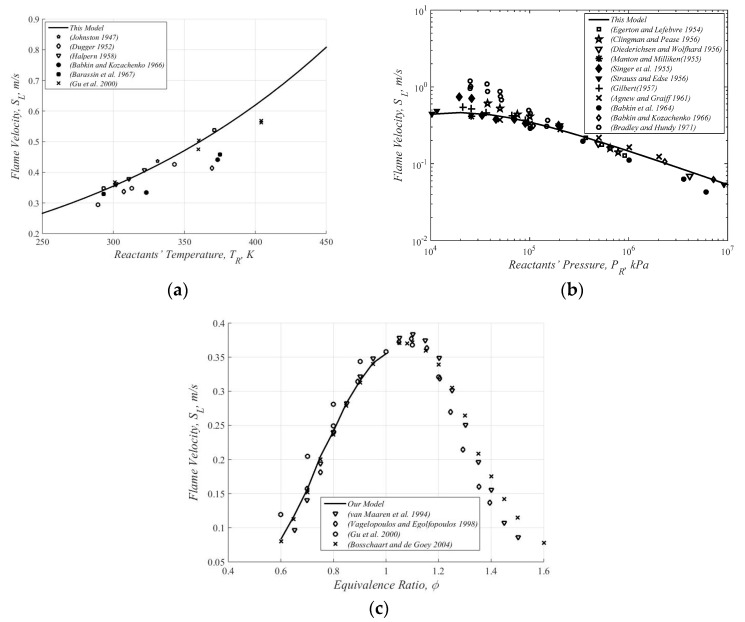
Some experimental results [5] and the present prediction (solid line) for methane/air mixtures. Here the equivalence ratio is defined as the ratio between the air-fuel mass ratio and the stoichiometric fuel-air mass ratio. (**a**) Flame velocity vs. reactants’ temperature [6,7,8,9,10,11]. (**b**) Flame velocity vs. reactants’ pressure [9,12,13,14,15,16,17,18,19,20,21]. (**c**) Flame velocity vs. equivalence ratio [11,22,23,24].

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
