# Peer review of "Implementation of the Onsager Theorem to Evaluate the Speed of the Deflagration Wave"

_entropy, 2020, doi:10.3390/e22091011_

Round 1
Reviewer 1 Report
Authors answered all of my questions and the manuscript may be published.
Authors state that they improve the slow flame model and now the momentum
equation is fulfilled. But they make a lot of other nonphysical assumptions
"While assuming a linear temperature profile and ignoring the mass diffusion"
(Page 3, line 108).
This are strong enough assumptions, and I ask the authors to think once more
Do the manuscript is worth to be published?
Reviewer 2 Report
1.-Use only “steady state”, it is not necessary steady state steady flow.
2.-Some reference are missing, for example:
Line 36: Rankine, Hugoniot and Rayleigh.
Line 41: Champman and Jouguet.
Line 47: Mallard nad Le Chatelier.
Line 50: Zeldovich, Frank-Kamenetskii and Semenov.
3.- Check all the references missing in the paper.
4.- Line 58: it is stated “...,we suggest to introduce the Onsager relationship [3],...” the sentence is not correct, Reference [3] is not yours.
5.- The logical assumptions that you used are oversimplified assumptions (no viscous, heat transfer and matter diffusivity).
6.- The novelty of your work is not clear. Why the values of the propagation flame deflagration velocity are better than the predicted values and experimental values of others works.
7.- What is the advance in the science with your propose to obtain the same results against others authors?
8.- What else can you do with your method than the others?
9.- What is the potential of your proposed method?
10.-In Figure 1. Why the entropy is decreasing in the detonation?
Reviewer 3 Report
Manuscript:
Implementation of the Onsager Theorem to Evaluate the Speed of the Deflagration Wave
Comments:
Abstract:
"... Quite a good fitting, similarly to the classic theory predictions, has been obtained"
(a somewhat vague formulation)
Line 145
"Assuming ? = ????? (here ? is the thermal conductivity of the gases ...)"
When and why can it be assumed that K = const?
K is known to be temperature-dependent. How much of a mistake does this assumption make?
Line 148
"Assuming an ideal gas with ?? = ?????, ..."
cp is temperature-dependent.
The explanation for that assumption is missing.
Line 184 ..Eq. 15 or 16
Line 185-187 and Eq. 16
"Eq. (16) suggests that the end results for the burning velocity doesn’t noticeably differ from the value as predicted by the classical theory."
noticeably differ - This should be clarified and show how much this deviation is.
Line 196
(? > 1 in Fig. 1C), are inadequate
- what is equivalence ratio? (not explained and defined in Nomenclature)
- fig 1C? - maybe Fig 2C?
For readers who are not close in this area would be welcome clarification of certain assumptions and estimates how much the error thus introduced into the calculation.
Round 2
Reviewer 2 Report
The manuscript is accepted.
This manuscript is a resubmission of an earlier submission. The following is a list of the peer review reports and author responses from that submission.
Round 1
Reviewer 1 Report
The study applied the Onsager Theorem to evaluate the Champman-Jouguet Deflagration Point. The suggestion of the introduction of the theorem should be well validated, experiments are needed. A detailed investigation should be carried out, it should consider all the phenomena to find velocity, species, temperature, etc. I must reject.
There are many flaws in the paper:
Line 13-15. It was stated that: “we suggest to introduce a simplified version of The Onsager relationship, which accounts for the entropy increase due to the heat transfer process from the front layer to its successive layer...”. However, results of the entropy are not shown in the paper.
-The introduction is not well address. An extensive review of the literature should be included (the sate of the art).
-Equations of continuity, momentum and energy are written in steady state and the entropy is written in unsteady state
-Results of the entropy are missing.
-Results of species are missing.
- The abbreviation “SSSF” should be defined.
- Nomenclature of all the variables should be included in the Nomenclature Section including the units.
-Reference are old, they should be updated.
-In the Figure 1b, the units of the pressure are incorrect.
-The advantages and disadvantages of your proposed model vs other models are missing.
Reviewer 2 Report
Article "Implementation of the Onsager Theorem to Evaluate
the Chapman-Jouguet Deflagration Point" by E. Sher, I. Moshkovich-Makarenko,
Y. Moshkovich and B. Cukurel is written so unclear that I am not able to make
a definite decision on possibility of its publication. I try to make several
comments that will help to make the manuscript more clear in my opinion.
1) Authors try to estimate the speed of one dimensional slow combustion wave
(deflagration) which has no any connection with "Chapman-Jouguet Deflagration".
I ask authors to change absolutely misleading title by something like this:
"Implementation of the Onsager Theorem to Evaluate the speed of slow combustion
wave."
2) What does it mean "SSSF conditions" (lines 25-26).
3) "It however can be shown that this classical solution doesn’t conform to
the momentum equation [1]" (lines 53,54). If it is shown in Ref[1] please give
the exact page and line, otherwise please show by yourself.
4) "Equation (7) provides the additional equation that is needed to solve Eqs.
(1-4) (or 5-6) for a deflagration wave." (line 107, 108). How exactly authors
propose to couple the differential equation (7) with algebraic system (1)-(4)?
They need to predict profile T(x)? And what about diffusive flukes of the fuel?
Why authors ignore the Lewis number, which impacts on the speed?
5) Where authors employ ideas from line 121 till line 132?
6) Eqs. 12,13 - what are fluxes and driven forces here? Where are the molecular
flux? How you choose L_{i,j}?
7) Eq. 14 - what is the origin of the right part?
8) "Following the classic thermal theory, in which the heat generation is
compared to the heat convection, the flame thickness can be easily expressed
in terms of the rate of heat generation which include an Arrhenius type
expression [1]" (lines 146-148). Please insert the equation and short
explanations.
9) What is "K" in Eq. (13)-(15)? How authors define its value?
10) Please also insert the flame speed according to Zeldovich, Frank-Kamenetskii
and Semenov theories to Fig. 1.